# Development of a Cell-Based SARS-CoV-2 Pseudovirus Neutralization Assay Using Imaging and Flow Cytometry Analysis

**DOI:** 10.3390/ijms241512332

**Published:** 2023-08-02

**Authors:** Jerilyn R. Izac, Edward J. Kwee, Linhua Tian, Elzafir Elsheikh, Adolfas K. Gaigalas, John T. Elliott, Lili Wang

**Affiliations:** Biosystems and Biomaterials Division, National Institute of Standards and Technology (NIST), Gaithersburg, MD 20899, USA; linhua.tian@nist.gov (L.T.); elzafir.elsheikh@nist.gov (E.E.); adolfas.gaigalas@nist.gov (A.K.G.); john.elliott@nist.gov (J.T.E.); lili.wang@nist.gov (L.W.)

**Keywords:** SARS-CoV-2, neutralization, pseudovirus, antibodies, diagnostic, flow cytometry, imaging

## Abstract

COVID-19 is an ongoing, global pandemic caused by the novel, highly infectious SARS-CoV-2 virus. Efforts to mitigate the effects of SARS-CoV-2, such as mass vaccination and development of monoclonal therapeutics, require precise measurements of correlative, functional neutralizing antibodies that block virus infection. The development of rapid, safe, and easy-to-use neutralization assays is essential for faster diagnosis and treatment. Here, we developed a vesicular stomatitis virus (VSV)-based neutralization assay with two readout methods, imaging and flow cytometry, that were capable of quantifying varying degrees of neutralization in patient serum samples. We tested two different spike-pseudoviruses and conducted a time-course assay at multiple multiplicities of infection (MOIs) to optimize the assay workflow. The results of this assay correlate with the results of previously developed serology and surrogate neutralization assays. The two pseudovirus readout methods produced similar values of 50% neutralization titer values. Harvest-free in situ readouts for live-cell imaging and high-throughput analysis results for flow cytometry can provide unique capabilities for fast evaluation of neutralization, which is critical for the mitigation of future pandemics.

## 1. Introduction

*Severe acute respiratory syndrome coronavirus 2* (SARS-CoV-2), the cause of coronavirus disease 2019 (COVID-19), emerged in December 2019 and resulted in a global pandemic [1]. SARS-CoV-2 displayed significant pathogenicity and has caused significant mortality worldwide. A variety of approaches to combat SARS-CoV-2 have resulted in several prophylactics and therapeutics, including RNA- and viral vector-based vaccines, new antivirals, and monoclonal antibodies (mAbs) [2]. SARS-CoV-2 uses a glycoprotein, called spike protein, to enter the host cell through the host cell receptor ACE-2 [3,4]. TMPRSS2, a serine protease, cleaves the spike protein and facilitates viral entry [5]. Broadly protective vaccines prevent the SARS-CoV-2 spike and new variants from binding to the ACE-2 and TMPRSS2 host cell receptors and are vital for combating the pandemic [6]. To predict the effectiveness of these vaccines, it is paramount to determine the titer of neutralizing antibodies (nAbs). 

Several methods to quantify nAb titers in patient serum have been established, including live virus, pseudovirus, and ELISA-based neutralization assays [7,8]. Use of live pathogenic SARS-CoV-2 requires biosafety level 3 (BSL-3) containment that is not available for most laboratories performing diagnoses of infection, development of antivirals, and other basic or applied research. Alternatively, neutralization assays based on pseudoviruses offer better safety and improved ease of use, only requiring BSL-2 containment. Pseudovirus assays are comparable with live pathogenic SARS-CoV-2 microneutralization assays when implemented as alternative assays [9,10]. Pseudoviruses are recombinant viruses that are engineered to express a surface protein from another virus (i.e., SARS-CoV-2 spike protein) used on the coat of the pseudovirus [11]. Genes within a pseudovirus are altered to limit or abolish native surface protein expression, and a plasmid is used to express alternative surface proteins and, sometimes, a fluorescent reporter. Numerous cell lines expressing ACE-2 and/or TMPRSS2 and pseudoviruses modified with SARS-CoV-2 spike protein, including lentiviral and vesicular stomatitis virus (VSV)-based pseudoviruses, have been generated to facilitate the study of SARS-CoV-2 [12,13]. In this study, we developed a pseudovirus neutralization assay using a VSV pseudovirus with a SARS-CoV-2 spike protein and a Green Fluorescent Protein (GFP) reporter, which served as a convenient reporter of infection. Neutralization was measured by both live-cell imaging and flow cytometry. We show that both readout methods quantified the presence and absence of neutralizing antibodies in patient serum samples. Live-cell imaging and flow cytometry analysis showed comparable quantifications of the neutralization. Comparison of the pseudovirus assay to the bead-based serology and neutralization assays showed high correlation, and receiver operator curve (ROC) analysis resulted in a high area under the curve (AUC).

## 2. Results

### 2.1. Pseudovirus Evaluation

Two different pseudoviruses were tested, a lentivirus and a VSV-based pseudovirus with the original Wuhan-Hu-1-strain spike protein and GFP reporter. After 24 h at a multiplicity of infection (MOI) of 1.0, the VSV-based pseudovirus had the most GFP expression with an average of 80.9 ± 3.06% compared to the lentivirus at 26.6 ± 1.99% (Figure 1). Additionally, the VSV-based pseudovirus was significantly brighter, which enabled better visualization for imaging and gating for flow cytometry. To improve both the speed and ease of use of the assay, the VSV-based pseudovirus was used for the neutralization assay. Additionally, no coating or fibronectin coating was tested with VSV-pseudovirus infections, and no changes in GFP expression were observed. 

### 2.2. Assay Time-Course Optimization

To develop a rapid and easy-to-use neutralization assay, cells were infected with the pseudovirus to determine the optimal time and MOI for the assay. HEK293T-hACE-2-TMPRSS2-mCherry cells were infected with the VSV-spike pseudovirus at MOIs of 0.5, 1.0, or 2.0 for 2, 4, 8, 16, and 24 h (Figure 2). Maximum GFP expression was reached at 8 h for all MOIs. A two-way ANOVA with multiple comparisons showed that the 16 h timepoints were not significantly different in GFP expression compared to the 8 h timepoint for each MOI (*p* = 0.067), and GFP expression at 16 h did not differ between the three MOIs tested (*p* = 0.299). However, the 16 h timepoint at an MOI of 0.5 was chosen for the neutralization assay because the difference between the timepoints and MOIs was insignificant. These assay parameters allowed for more manageable workflow timing and reduced the virus consumption. A small increase in dead cells was seen at the 24 h timepoint at an MOI of 2.0.

### 2.3. Serum Evaluation

Following the addition of the virus and serum mixtures to the cells, each well was imaged after 16 h (Appendix A). The resulting mCherry and GFP images were segmented and analyzed to determine the percent neutralization of each dilution (Appendix A). The cells were dissociated with trypsin EDTA from the plate and analyzed using flow cytometry. Each dilution was also analyzed using FlowJo, and the data were analyzed using the same gating strategy (Appendix A). Using both imaging and flow cytometry readouts for the pseudovirus assay, no neutralization was identified for any of the 28 known negative serum samples. Neutralization was identified in 44 out of 50 known positive serum samples by calculating the NT50 from the calculated neutralizations across the serum dilutions (Figure 3A). Six serum samples that were known positive samples, but had low IgG titers, were not identified for neutralization by either neutralization assay. An additional six serum samples did not produce an NT50 value for the surrogate bead-based neutralization assay. For comparison of NT50 values in the positive serum samples measured using imaging and flow analysis, a Wilcoxon matched-pairs signed-rank test was performed. No significant differences were found between the imaging and flow analysis (*p* = 0.446), and a Spearman rank correlation test showed high correlation between the two assays. (r_s_= 0.988, *p* value < 0.0001) (Figure 3B).

### 2.4. Bead-Based Surrogate and Serology Assay Comparison

The NT50 values were averaged between the live-cell imaging and the flow cytometry readouts to determine average pseudovirus NT50 values. These were then compared to two bead-based assays [14], a previously described spike IgG serology assay and a bead-based surrogate neutralization assay. Evaluation using a Spearman rank correlation coefficient (r_s_) showed significant correlation between the NT50 values of the pseudovirus assay and these assays (serology: r_s_ = 0.797, *p* < 0.0001, and surrogate: r_s_ = 0.880 *p* < 0.0001, Figure 4). While all samples provided an IgG titer for the serology assay, only samples with an NT50 value were compared to the pseudovirus assay. Next, all assays were compared to one another using ROC analysis, which determined that the serology assay had the highest AUC (0.965, *p* < 0.0001) compared to the pseudovirus (0.946, *p* < 0.0001) and surrogate (0.902, *p* < 0.0001) assays (Figure 5). Based on the AUCs, all three assays were excellent predictors for the presence of neutralizing antibodies; however, the serology assay performed the best. The serology and pseudovirus assays had almost identical sensitivities and specificities, while the surrogate neutralization assay had lower values at the optimal thresholds. With an optimal threshold of 16.28 BAU/mL, the sensitivity of the serology assay was 89.58 and the specificity was 100. For the pseudovirus assay, the optimal threshold was an NT50 of 2.006 with a sensitivity of 89.8 and a specificity of 100%. The surrogate assay had an optimal threshold NT50 value of 8.87, with a sensitivity of 77.6% and a specificity of 82.8%. 

## 3. Discussion

One goal for pandemic preparedness is to develop rapid, easy-to-use, high-throughput assays to help determine COVID-19 serological diagnosis and vaccination efficacy. Pseudovirus assays offer BSL-2 convenience and ease of use to laboratories measuring serology neutralization. When testing two pseudoviruses, we found that the VSV-based pseudovirus resulted in a higher percent GFP expression after 24 h than the lentivirus-based pseudovirus (Figure 1). After selecting the VSV pseudovirus, a time-course assay tested three different MOIs to determine the optimal time and MOI to use for this assay (Figure 2). Maximum infection, which was considered the highest percent GFP expression of all the cells, occurred around the 8 h timepoint for MOIs of 1.0 and 2.0. Maximum infection at the MOI of 0.5 happened around the 16 h timepoint and split the setup and endpoint cell processing into two days. Since the 16 h timepoint did not statistically differ in percent GFP expression from 8 h (*p* = 0.069) and the three MOIs at 16 h did not statistically differ (*p* = 0.299), the MOI of 0.5 and the 16 h timepoint were chosen.

When considering the pseudovirus assay readout methods, both live-cell imaging and flow cytometry readouts provided comparable measurements of 50% neutralization titer. Using a dual approach for determining the neutralization provided insight on optimizing samples for both flow cytometry and imaging. The mCherry reporter in the cell line was useful for the imaging readout by providing a fluorescence-based method to segment and normalize the total cells to the virus-infected cells that expressed GFP. Analysis of the total cells by imaging required the optimization of the mCherry segmentation parameters specific to this cell type, while the flow cytometry analysis based on scatter was more straightforward (Appendix A). The cell count and distribution within the well were important for the optimization of the imaging assay readout. Optimization of the cell plating density was required to balance sufficient cells needed to yield sufficient sampling using flow cytometry while not reaching confluency when the image segmentation was challenging. The plating density was ultimately optimized at 7500 cells/well for this assay. Imaging also revealed that cells were frequently lost during media exchanges when the cells were plated directly on tissue culture plastic. Fibronectin improved the HEK293 cell adhesion and caused fewer cells to wash away during media exchanges, which increased the cell retention for the assay and the cell visualization for imaging [15]. Imaging, however, was sensitive to assay artifacts such as bubbles, requiring their removal, which sometimes resulted in fields of view being rejected for analysis. When considering the sample analysis, the imaging assay imaged the well plates directly with automated analysis output and without any sample harvest or preparation. Little sample preparation, which consisted of a sample harvest and two washes, was required for running the samples through the NxT Attune autosampler, forming both the flow cytometry and imaging high-throughput methods.

Overall, the imaging method for this neutralization assay relied on a fluorescent reporter within the cell, a fibronectin coating, and care to ensure that no bubbles were introduced during the addition of the serum–virus mixture. The flow cytometry method required additional sample harvests and washes but had a straightforward gating strategy to quantify neutralization. The imaging method provided automated in situ cell measurements within the plate and automated image analysis, but the flow cytometry method was able to process more cells than were able to be imaged. Nine 10× magnification fields of view were used to image cells to avoid microplate well-edge effects. This area represented approximately 13% of the total well surface area and sampling of the cell population, whereas flow cytometry was able to analyze the entire cell population. Despite the smaller sampling by imaging, there were no significant differences between the NT50 values of the imaging and flow cytometry analyses (*p* = 0.446), and the results were highly correlated, demonstrating potential to use these analysis methods interchangeably (Figure 3B, r_s_ = 0.988). Additionally, these two orthogonal measurements of neutralization on the same pseudovirus-infected samples provided assurance that the measured percent infectivity and resulting pseudovirus neutralization titer was correct.

Serum samples processed via the pseudovirus neutralization assay were concurrently processed with the previously validated bead-based serology assay and bead-based surrogate neutralization assay [14]. Comparisons of the pseudovirus assay to the bead-based serology and neutralization assays demonstrated significant correlations to both assays (Figure 4, r_s_ = 0.7971 and r_s_ = 0.8796 *p* < 0.0001). Higher anti-spike IgG titers correlated with higher NT50 values, indicating that the pseudovirus neutralization assay has potential to be benchmarked to a quantitative value (BAU/mL), enabling improved comparability of NT50 titers. Additionally, differences between the characteristics of these assays could result in low correlation for some of the samples. First, the pseudovirus assay was a cell-based assay, while the serology and surrogate assays were both bead-based assays. For the bead-based neutralization assay, the correlation showed that the NT50 values had a trend of being higher for the pseudovirus assay than the surrogate assay. One potential explanation for this was that the target antigen for the surrogate assay was the RBD protein compared to the pseudovirus containing a spike protein. It is likely that there are some neutralizing antibodies directed towards other regions of the spike not within the RBD domain [7,8]. There were also differences in the neutralization output metrics between the pseudovirus assay, which output NT50, and the serology assay, which output BAU/mL. When considering the false negative rate, the pseudovirus neutralization assay was not able to identify six known positive serum samples. Of these samples, the serology assay only considered two as positive. The surrogate neutralization assay was not able to identify twelve confirmed infection serum samples as positive, including the same six samples as the pseudovirus assay. It is possible that the some of these false negative samples were sampled from patients early during infection and had little to no IgG antibodies or neutralizing antibodies directed to the RBD or the overall spike. Despite the differences in performance between the three assays, a high level of correlation was still observed when comparing the pseudovirus assay to the two established bead-based assays (Figure 4), thereby providing validation of the pseudovirus assay results. Another potential explanation for the differences in neutralization could be infection by different SARS-CoV-2 variants, such as Omicron, Delta, Gamma, or Alpha. A potential limitation of the study is that samples for this study were collected prior to knowledge of other variants than Wuhan-Hu-1 or the development of methods that can identify different variants. Differences in neutralization based on infections by different variants will be addressed in future studies. 

In summary, this work highlights a cell-based pseudovirus neutralization assay that is straightforward and easy to perform. The use of a pseudovirus offers a safer alternative to the live-virus neutralization assays, which require BSL-3 laboratories. The assay can be easily adapted using pseudoviruses of new SARS-CoV-2 variants and future pandemic viruses. The hands-off nature of the imaging readout and the high-throughput capability of the flow cytometry readout show fast and convenient ways to quantify neutralization, which is critical to future pandemic preparedness. The methods demonstrated here can serve an important role to quantify neutralizing antibody titers needed for the future development of antivirals and monoclonal therapeutics.

## 4. Materials and Methods

### 4.1. Cell Growth

HEK293T (ATCC, CRL-3216), HEK293T-hAce2 (BEI Resources, NR-52511), and HEK293T-hAce2-TMPRSS2-mCherry (BEI Resources, NR-55293) cells were grown in Eagle’s Minimum Essential Medium (EMEM, Corning 10-009-CV) + 10% heat-inactivated fetal bovine serum (HI-FBS) (Gibco, Billings, MT, USA, 16140071). All cells were grown with vendor-specified selection antibiotics in a T25 flask to 70–90% confluency after 2 to 3 days and were subsequently passaged. The cells were collected using Trypsin EDTA (Corning, 25-053-CI) and seeded at 1 million cells per flask. Cells were used between passages 3 and 20.

### 4.2. Pseudovirus Evaluation 

To improve cell adhesion, 96-well tissue-culture-treated plates (Corning, Corning, NY, USA, 3595) were coated with 10 μg/mL of fibronectin (Sigma, St. Louis, MO, USA, F1141) for 4–6 h before washing and seeding the cells. HEK293T-hAce2-TMPRRS2-mCherry cells were seeded at 7500 cells per well and allowed to adhere at 37 °C with a 5% volume fraction of CO_2_ for 24 h. HEK293T cells were used as a negative control. After 24 h, the average cell count for each well was determined after trypsinizing 10 wells on each plate using a Multisizer 3 Coulter Counter (Beckman Coulter, Brea, CA, USA). To calculate a multiplicity of infection (MOI) of 1.0 for SARS-CoV2, SΔG-GFP Pseudotyped VSV (Creative Biogene, Shirley, NY, USA CoV-012) with a provided virus titer of 10^7^ plaque-forming units per mL was used. For SARS-CoV-2 spike pseudotyped lentivirus with an eGFP reporter (AMSBIO, Cambridge, MA, USA, AMS.79981), a virus titer of 2 × 10^5^ transducing units per mL was provided by the manufacturer and used to calculate the MOI. Both pseudoviruses had SARS-CoV-2 spike proteins from the Wuhan-Hu-1 strain. The MOI was calculated by multiplying the average cell count by the desired MOI and dividing by the virus titer provided by the manufacturer. The infection proceeded for 24 h before supernatant was collected, and attached cells were washed and trypsinized for 3–5 min. Trypsin was inactivated with media, and the cells and supernatant were combined and subsequently washed and run through the NxT Attune flow cytometer (ThermoFisher, Waltham, MA, USA).

### 4.3. Assay Time-Course Optimization 

HEK293T-hACE-2-TMPRSS2-mCherry cells were plated on a 96-well tissue-culture-treated plate at 7500 cells per well, placed in a 37 °C incubator with a 5% volume fraction of CO_2_ for 24 h, and tested in the time-course study with three different MOIs of VSV-based pseudovirus: 0.5, 1.0, and 2.0. The MOI was calculated by trypsinizing the cells as described in Section 2.2, and the GFP expression was monitored by flow cytometry at 0, 2, 4, 8, 16, and 24 h. After each timepoint, supernatant and cells were harvested, as in Section 2.2. Cells were stained for Live Dead Violet (ThermoFisher, L34963) per the manufacturer’s recommendations.

### 4.4. Pseudovirus Neutralization Assay

Serum samples of convalescent, vaccinated, and pre-COVID patients were collected before 2021 and provided by the Frederick National Laboratory for Cancer Research of the National Cancer Institute, the Center for Disease Control, and Abbott Laboratories through respective material transfer agreements (MTAs). Seventy-seven serum samples were tested, including twenty-eight known negative samples (Appendix A). These serum samples were used for all assays described in this study to make a direct comparison of the three assays. The sample identity was blinded until after analysis. Known neutralizing and non-neutralizing monoclonal antibodies were provided by Regeneron through an MTA and were used in the study as positive and negative controls on each plate. The World Health Organization international standard (WHO IS, NIBSC code: 20/136) was also used in the study as a control [7]. 

As described in Section 2.2, 96-well tissue-culture-treated plates were coated with fibronectin for 4–6 h. The wells were seeded with 7500 HEK293T-hACE2-TMPRSS2-mCherry cells per well and allowed to adhere at 37 °C with a 5% volume fraction of CO_2_ for 24 h. After 24 h of adhesion, an average cell count from 10 wells was used to calculate the MOI.

A 9-point curve to obtain 50% neutralization titer (NT50) was generated with an initial 5-fold dilution of serum in OPTI-MEM (Gibco, 31985070) + 2% HI-FBS, and then 8 subsequent 3-fold dilutions (Appendix A). VSV-based pseudovirus expressing SARS-CoV-2 spike protein from SARS-CoV-2 (Wuhan-Hu-1) with a GFP reporter (Creative Biogene, CoV-012) was added to a media-diluted serum for a final MOI of 0.5 and incubated for 1 h at 37 °C. Media were removed from the HEK293T-hAce2-TMPRSS2-mCherry cells, and the virus–serum mixture was added to the wells and immediately placed on a Cytation 5 Cell Imaging Multimode Reader (Agilent, Santa Clara, CA, USA) for live-cell imaging for 16 h. After imaging, samples were harvested and run on the NxT Attune flow cytometer. Controls included media-only wells, virus and media wells, a known neutralizing mAb, and a non-neutralizing mAb. The known non-neutralizing mAb was diluted at 6 μg/mL and 0.06 μg/mL in singlet, while the neutralizing mAb was tested in singlet with 5 ten-fold serial dilutions starting at 6 μg/mL.

### 4.5. Live-Cell Imaging Data Processing and Analysis

Using the Cytation 5, brightfield, GFP, and mCherry fluorescence images were acquired using a 10x objective across nine fields of view per well. The nine fields of view were acquired in the center of the well to avoid well-edge effects. Imaging was performed every 75 min across all sample wells. Image processing and analysis were performed using a custom script implemented with MATLAB R2022a (MathWorks, Natick, MA, USA). GFP and mCherry images were segmented using the empirical gradient threshold method [16]. The total cell area and infected cell area were determined from the mCherry and GFP segmented images, respectively, enabling calculation of the percent GFP-positive cells at every timepoint (Appendix A). The percent GFP-positive cells over the time course was fit to a logistic curve. The maximum of the logistic curve was used to determine the endpoint percent GFP for a given dilution.

### 4.6. Serum Sample Analysis via Flow Cytometry

After imaging, supernatants from each well were collected and wells were washed with phosphate-buffered saline (PBS). For each wash, supernatants were collected. Wells were trypsinized for 3 to 5 min, the trypsin was inactivated with media, and each sample was diluted with PBS + 2% HI-FBS. Cells were spun at 200× *g* for 5 min and supernatants were removed. Cells were resuspended in PBS + 2% HI-FBS and put on a non-tissue-culture-treated 96-well plate for loading on the NxT Attune Autosampler. To set up fluorescence compensation, HEK293T-hAce2 cells were used as an unstained control. HEK293T-hAce2 cells that had been infected with VSV-ΔG spike pseudovirus for 24 h were used for GFP compensation, while HEK293T-hAce2-TMPRSS2-mCherry cells were used for mCherry compensation. Samples were analyzed using FlowJo v10.8.1 (Becton Dickinson, Sparks, MD, USA) and gated for cells based on the side and forward scatter areas, single cells based on the forward scatter height and area, and GFP expression (Appendix A). The GFP gate was determined based on the autofluorescence of the non-infected cell control. The same gates were applied to all samples to determine the percent of GFP-infected cells. Only wells with a minimum of 1500 single cells were included in the analysis. 

The spike IgG serology assay and surrogate neutralization assay were performed as previously described [14]. Briefly, spike (Wuhan-Hu-1 strain)-conjugated MagPlex-C microbeads were aliquoted onto a 96-well plate at 10,000 beads/well. Patient serum samples, diluted controls, and reference standards were serially diluted and incubated for 30 min in the dark at room temperature while shaking (800 rpm). A magnetic separator was used to pull beads down for at least 1 min on the 96-well plate, and then the wells were washed with 1X PBS, 1% bovine serum albumin (BSA), and 0.05% Tween 20 (PB-T) in triplicate. PE-labeled anti-IgG was added and incubated again. After two washes, 100 μL of wash buffer was applied and samples were analyzed with 3000 to 5000 gated bead events on the CytoFLEX LX (Beckman Coulter, CA, USA) flow cytometer. The spike IgG serology assay was standardized to the WHO IS for anti-SARS-CoV-2 immunoglobulin (NIBSC code: 20/136), and the results are presented in binding antibody units per milliliter (BAU/mL). 

For the bead-based surrogate neutralization assay, patient serum samples were serially diluted and incubated with RBD (Wuhan-Hu-1 strain)-coated beads in a 96-well plate at 10,000 beads per well for 30 min, shaking at 750 rpm at RT in the dark. The plates were washed three times with PB-T after using a magnetic separator for 1 min. Biotinylated ACE-2 (0.625 µg/mL) was added to each well and incubated while shaking for 1 h. The beads were washed again and then incubated with PE-Streptavidin (0.6 µg/mL) for 30 min of shaking. The beads were washed and resuspended in PB-T and run on the Cytoflex S (Beckman Coulter, Brea, CA, USA) flow cytometer. If neutralization occurred, lower serum dilutions resulted in lower fluorescence signals.

### 4.7. Statistical Analysis

All statistical analyses were conducted in GraphPad Prism 9 (GraphPad Software, San Diego, CA, USA).

For the time-course assay, a two-way analysis of variance (ANOVA) with multiple comparisons was used to determine if there was a statistically significant difference between the three MOIs tested across the different timepoints.

For the pseudovirus neutralization assay, the percent GFP was used to calculate the percent reduction. For the flow cytometry analysis method, the reduction percentage was defined as Reduction% = 100 × ((%_GFP_)_Max_ − %_GFP_)/(%_GFP_)_Max_), where (%_GFP_)_max_ is the control sample treated with pseudovirus without a serum sample. For the imaging analysis method, the reduction percentage was defined as Reduction% = 100 × ((Area%_GFP_)_max_ − Area%_GFP_)/(Area%_GFP_)_Max_), where (Area%_GFP_)_max_ is the control sample treated only with pseudovirus without a serum sample. The percent reduction was determined for each dilution. The NT50 was calculated using the variable slope model based on the Hill slope of the dilutions, and the NT50 was determined for both readout methods [14]. A Wilcoxon matched-pairs signed-rank test and a Spearman rank correlation test were used as nonparametric tests to compare the NT50 values from the imaging and flow cytometry analysis to one another [17,18]. Six samples with known COVID infection but low IgG titers did not give an NT50 value for the pseudovirus assay, and twelve samples did not give an NT50 value for the bead-based surrogate assay. Only samples that gave an NT50 value were used to determine correlation, with negative serum samples not included either.

A receiver operating characteristic (ROC) curve was generated, including all samples for the three assays, to determine the sensitivity and specificity [19].

## Figures and Tables

**Figure 1 ijms-24-12332-f001:**
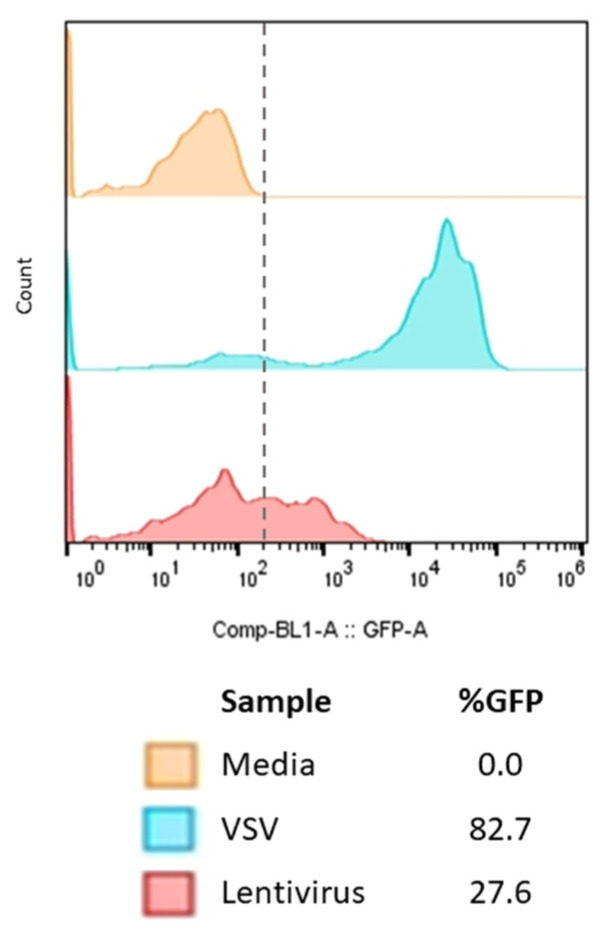
Testing Two Different Spike-based Pseudoviruses for GFP Expression. We tested two commercial spike-VSV and spike lentivirus-based pseudoviruses with GFP and eGFP reporters, respectively. An MOI of 1 for each pseudovirus was applied to HEK293T-hACE-2-TMPRSS2-mCherry cells and incubated for 24 h. After 24 h, the cells were washed and processed on an NxT Attune Flow Cytometer. Representative data are shown from one sample of two technical replicates with three biological replicates each. An average of GFP expression of 80.9 ± 3.06% was calculated for VSV and an average of 26.6 ± 1.99% for lentivirus. The dashed line indicates what is considered background, which was based on media only controls.

**Figure 2 ijms-24-12332-f002:**
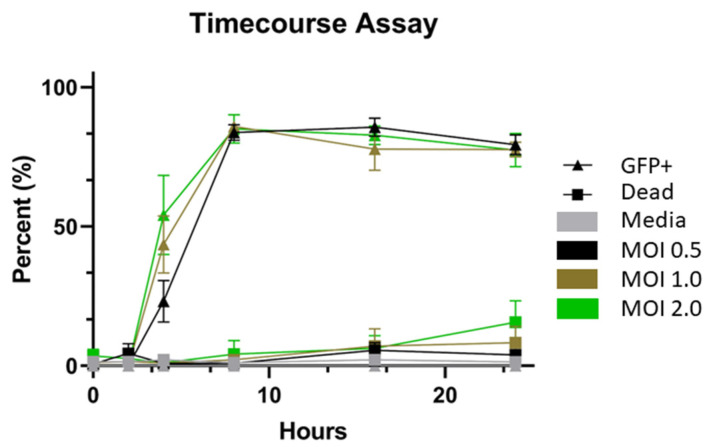
Time-course Assay Examining HEK293T-hACE-2-TMPRSS2-mCherry Cells after VSV-spike Pseudovirus Incubation at Multiple MOIs. Cells were stained with live/dead violet staining before processing on an NxT Attune Flow Cytometer. The average and standard deviations of 2 biological replicates with 3 technical replicates each (*n* = 6) are shown with error bars. The results of a two-way ANOVA with multiple comparisons showed that there was no statistically significant difference in infection or cell death between the 8 and 16 h timepoints and that there was no significant difference between the three MOIs at the 8 and 16 h timepoints as well. Based on these results, reduced pseudovirus consumption, and ease of use, an MOI of 0.5 and the 16 h timepoint were chosen for the neutralization assay.

**Figure 3 ijms-24-12332-f003:**
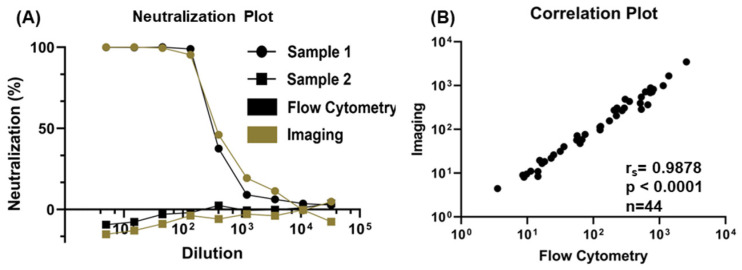
Neutralization 50 (NT50) Comparison of Two Analysis Methods, Imaging and Flow Cytometry. (**A**) Neutralization from an initial five-fold dilution followed by eight consecutive dilutions of both a positive (circle) and a negative (square) serum sample are shown. Percent neutralization for each dilution was calculated by normalizing the percent of infected cells to a virus-only control sample. The percent neutralization was used to determine the NT50 for each sample. Neutralizations from both methods were comparable. A correlation plot showing NT50 results from imaging compared to flow cytometry results (**B**). A Spearman rank correlation test demonstrated high correlation between the imaging and flow cytometry readouts.

**Figure 4 ijms-24-12332-f004:**
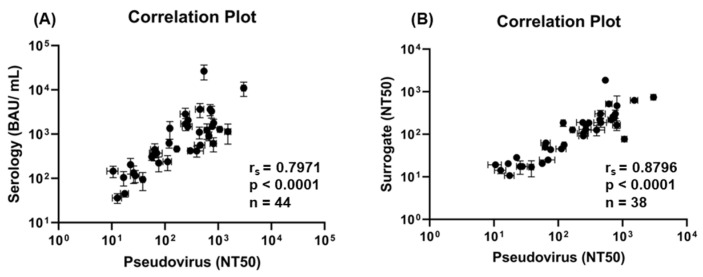
Correlations of Pseudovirus Assay with Serology and Surrogate Assays. Serology assay results (BAU/mL) were plotted against pseudovirus NT50 results (**A**). NT50 values were compared for pseudovirus and surrogate assay results (**B**). Both comparisons demonstrated high correlation, as shown by the high correlation coefficients and statistical significance calculated by the Spearman rank correlation test. The pseudovirus assay showed higher correlation with the surrogate assay than the serology assay. Due to differences in that some samples were positive for neutralization under some assays but not others, different sample numbers (*n*) are shown in the two correlation plots.

**Figure 5 ijms-24-12332-f005:**
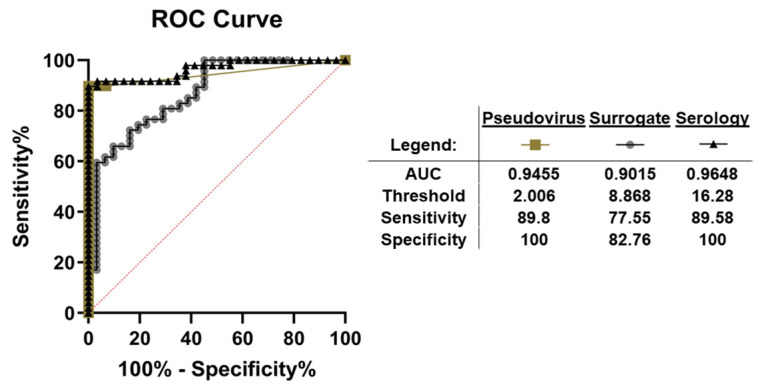
Receiver Operator Characteristic (ROC) Curve Analysis of Serology, Surrogate, and Pseudovirus Neutralization Assays. Based on the sample results, ROC curves were generated for the pseudovirus neutralization assay, bead-based surrogate assay, and spike IgG serology assay. Sensitivity and specificity were determined based on the specified threshold. Overall, the pseudovirus assay and serology assay had comparable areas under the curve (AUCs), sensitivity, and specificity.

## Data Availability

The data presented in this study are available on request from the corresponding author. The data are not publicly available due to privacy.

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
