# Peer review of "Development of a Cell-Based SARS-CoV-2 Pseudovirus Neutralization Assay Using Imaging and Flow Cytometry Analysis"

_ijms, 2023, doi:10.3390/ijms241512332_

Round 1
Reviewer 1 Report
I find this work very interesting and it describes a new cell based neutralization assay to be used to estimate the titer of virus neutralizing antibodies present in patients serum. I would like the authors to enrich the characteristics of the pseudovirus used with details and to know if the viral titer has been quantified. Also are the authors sure of the number of cells plated in the wells for the assay? After 24h from seeding do they get enough numbers for the infection?I think that the methods session could be improved with more details before publication.
Author Response
Point 1:
I find this work very interesting, and it describes a new cell based neutralization assay to be used to estimate the titer of virus neutralizing antibodies present in patients serum.
The authors thank the reviewer for all the comments made.
Point 2:
I would like the authors to enrich the characteristics of the pseudovirus used with details and to know if the viral titer has been quantified.
The virus titer was quantified by the manufacturers and MOI was based on the provided titer, this point was clarified in Section 4.2 lines 281-286.
Point 3:
Also are the authors sure of the number of cells plated in the wells for the assay?
Yes, the authors know the number of cells based on an average cell count, which is performed right before the virus is added. Edited in section 4.2 line 278 for clarification.
Point 4:
After 24h from seeding do they get enough numbers for the infection?
Yes, the authors get enough cells after seeding for 24 hours since this HEK293T-Ace-2-TMPRSS2 cell line has a doubling time of ~20 hours.
Point 5
I think that the methods session could be improved with more details before publication.
Thank you for your previous comments. By addressing these and the other reviewer’s comments, more details were provided in the method section to clarify for the reader.
Reviewer 2 Report
This study developed rapid, safe, and easy-to-use neutralization assays for COVID-19, which proved to be highly valuable.
SARS-CoV-2 is a virus known for its frequent mutations, and up to the emergence of Omicron, various mutations have been observed, leading to diverse interactions with serum. The method developed in this study is based on the early genomic sequence of the spike protein of the virus, and it may not fully account for all the mutations that have occurred. Therefore, while we have developed a simple and fast neutralizing antibody test, I must emphasize that its results may not fully reflect the changes in antibody levels or effects due to the presence of specific variants.
Author Response
Reviewer 2
Point 1
This study developed rapid, safe, and easy-to-use neutralization assays for COVID-19, which proved to be highly valuable.
SARS-CoV-2 is a virus known for its frequent mutations, and up to the emergence of Omicron, various mutations have been observed, leading to diverse interactions with serum. The method developed in this study is based on the early genomic sequence of the spike protein of the virus, and it may not fully account for all the mutations that have occurred. Therefore, while we have developed a simple and fast neutralizing antibody test, I must emphasize that its results may not fully reflect the changes in antibody levels or effects due to the presence of specific variants.
This comment has been addressed as per other reviewer comments (See supplemental table 2). At the time of sample collection, the original Wuhan variant was dominant among the population. We the authors agree that this a limitation of this study; however, future studies implementing this assay can be conducted to assess the difference among different variants.
Reviewer 3 Report
Main point:
I would like the authors to explain in more detail how the definition of neutralizing antibodies should change in light of the emergence of new variants of the virus, please discuss.
As a technical note, this article is valuable, however, over the past three years, many different assays based on pseudovirus technology have been developed. Due to the proteins used to develop this assay, the methods may be outdated as they do not provide useful information on neutralizing abilities against existing virus variants.
Minor point:
In line 66, the abbreviation MOI should be deciphered
Also please explain in more detail how you calculated MOI (section 2.2)
Please explain the origin of neutralizing and non-neutralizing monoclonal antibodies (section 4.4.), against which virus variant they were evaluated.
Please provide characteristics of the samples used to evaluate the tests in supplementary materials (they can be grouped by antibody titer and origin (vaccination or infection, or both).
Please explain to which virus variant S and RBD proteins used for serological assessment belong (section 4.6.).
Author Response
Reviewer 3
Main point:
Point 1:
I would like the authors to explain in more detail how the definition of neutralizing antibodies should change in light of the emergence of new variants of the virus, please discuss.
We appreciate this feedback and have added this information in the discussion (Section 3 Lines 249-254).
Point 2:
As a technical note, this article is valuable, however, over the past three years, many different assays based on pseudovirus technology have been developed. Due to the proteins used to develop this assay, the methods may be outdated as they do not provide useful information on neutralizing abilities against existing virus variants.
We thank the reviewer for bringing this subject up and we want to clarify that the infection samples were collected in the year 2020 now in line 301-302) and therefore, the most relevant variant was Wuhan-Hu-1. The vaccinated patients were also immunized against the original Wuhan strain. We clarify this point in the new supplemental table that describes the samples in more detail. We recognize that not including additional variant related samples in this study could be a potential limitation and is a future direction. We also want to mention that many variants manufacturers offer pseudoviruses to test additional variants.
Minor point:
Point 3:
In line 66, the abbreviation MOI should be deciphered
The authors agree and have added to line 67.
Point 4:
Also please explain in more detail how you calculated MOI (section 2.2)
The authors agree and have added in Section 4.2, lines 281-287.
Point 5:
Please explain the origin of neutralizing and non-neutralizing monoclonal antibodies (section 4.4.), against which virus variant they were evaluated.
We agree and added this information to section 4.4 lines 307-309.
Point 6:
Please provide characteristics of the samples used to evaluate the tests in supplementary materials (they can be grouped by antibody titer and origin (vaccination or infection, or both).
The authors agree and have created Supplemental Table 1 to provide more description of characteristics of samples, as suggested.
Point 7:
Please explain to which virus variant S and RBD proteins used for serological assessment belong (section 4.6.).
We agree and have added this information to lines 359 and 372 for the serology and surrogate neutralization assays, respectively.
Round 2
Reviewer 3 Report
I have no further comments